

# Elevation of MPF and MAPK gene expression, GSH content and mitochondrial distribution quality induced by melatonin promotes porcine oocyte maturation and development in vitro

Zimo Zhao[1,2,*], Ling Yang[2,*], Dan Zhang[3], Zi Zheng[1], Ning Li[1], Qianjun Li[1] and Maosheng Cui[1]

[1] Institute of Animal Science and Veterinary of Tianjin, TianJin, China
[2] College of Life Sciences and Food Engineering, Hebei University of Engineering, HanDan, China
[3] Tianjin Animal Disease Prevention and Control Center, TianJin, China
[*] These authors contributed equally to this work.

Corresponding author
Maosheng Cui,
tjxmcui2014@126.com

## ABSTRACT

The MPF and MAPK genes play crucial roles during oocyte maturation processes. However, the pattern of MPF and MAPK gene expression induced by melatonin (MT) and its correlation to oocyte maturation quality during the process of porcine oocyte maturation in vitro remains unexplored. To unravel it, in this study, we cultured the porcine oocytes in maturation medium supplemented with 0, $10^{-6}$, $10^{-9}$, and $10^{-12}$ mol/L melatonin. Later, we analyzed the MPF and MAPK gene expression levels by RT-PCR and determined the maturation index (survival and maturation rate of oocytes). The GSH content in the single oocyte, and cytoplasmic mitochondrial maturation distribution after porcine oocyte maturation in vitro was also evaluated. We also assessed the effects of these changes on parthenogenetic embryonic developmental potential. The oocytes cultured with $10^{-9}$ mol/L melatonin concentration showed higher oocyte maturation rate, and MPF and MAPK genes expression levels along with better mitochondrial distribution than the 0, $10^{-6}$, and $10^{-12}$ mol/L melatonin concentrations ($p < 0.05$). No significant difference was observed in the survival rates when the oocytes were cultured with different melatonin concentrations. The expression of the MPF gene in the oocytes cultured with $10^{-6}$ mol/L melatonin was higher than with $10^{-12}$ and 0 mol/L melatonin, and the expression of the MAPK gene in $10^{-6}$ and $10^{-12}$ group was higher than the control ($p < 0.05$). As far as the embryonic developmental potential is concerned, the cleavage and blastocyst rate of oocytes cultured with $10^{-6}$ and $10^{-9}$ mol/L melatonin was significantly higher than the $10^{-12}$ mol/L melatonin and control. In conclusion, $10^{-9}$–$10^{-6}$ mol/L melatonin significantly induced the MPF and MAPK gene expression; besides, it could also be correlated with GSH content of single oocyte, mitochondrial maturation distribution, and the first polar body expulsion. These changes were also found to be associated with parthenogenetic embryo developmental potential in vitro.

## INTRODUCTION

Oocyte maturation is an extensively used experimental technique in animal reproduction biotechnology. It involves the immature oocyte extraction from the ovaries, and it's in vitro culture until it has reached the maturation stage (*Hoelker et al., 2017*; *Lonergan & Fair, 2016*). During in vitro culture and maturation, oocytes generate a high amount of Reactive Oxygen Species (ROS) as a result of mechanical treatment, air, light, and other factors (*Waiz et al., 2016*). Although the physiological dose of ROS is favorable for the oocyte maturation and development, excessive ROS leads to adverse effects due to the DNA damage, mitochondrial dysfunction, lipid peroxidation, abnormal protein modification, and so on (*Rajani et al., 2012*).

Melatonin is an antioxidant, which binds and inhibits the oxygen free radical. Also, it serves as a protective barrier against the oxidative stress damage during oocyte maturation and development primarily by enhancing the intracellular glutathione level (*Agarwal & Majzoub, 2017*; *Gao et al., 2012*; *He et al., 2019*; *Tamura et al., 2008*). Glutathione is a crucial factor in the maturation and development of the oocyte as it eliminates the intracellular ROS (*Lee et al., 2016*). Mitochondria are ATP production sites, which are evenly distributed and diffused across the cytoplasm during the formation of oocytes with a high maturation quality (*Cui et al., 2009*). Mitochondria lead to energy production during the oocytes maturation and embryo development, and hence the mitochondrial distribution is correlated with oocyte maturation quality (*Babayev & Seli, 2015*), as well as with the oocyte fertilization and early embryo development (*Chappel, 2013*).

In the oocytes meiotic maturation, MAPK performs a vital role in the early embryo development processes, such as the initiation of GV, the promotion of nuclear maturation, and the oocytes maintenance at the MII stage (*Tripathi, Kumar & Chaube, 2010*). MPF, a cyclin B1, and cyclin-dependent kinase complex play a crucial in the meiosis maturation of oocytes. It maintains the normal meiosis cycle of oocytes and the normal cleavage of early embryos (*Baek et al., 2017*; *López-Cardona et al., 2017*).

Previous studies have comprehensively reported the roles of melatonin in the maturation and embryonic development of bovine oocytes (*Liang et al., 2017*), sheep oocytes (*Xiao et al., 2019*), porcine oocytes (*Li et al., 2015*) and mice oocytes (*Nikmard et al., 2016*). However, the melatonin-induced MPF and MAPK gene expression and its correlation with the oocyte maturation quality during porcine oocyte maturation in vitro, remains obscure. In-line with our previous findings, we hypothesize that MPF and MAPK genes expression also play a crucial role in the effect of melatonin promoting oocyte maturation quality in vitro. In the current study for the first time, we reported the expression patterns of MPF and MAPK genes induced by 0, $10^{-6}$, $10^{-9}$ and $10^{-12}$ mol/L melatonin and its association with oocytes maturation index during porcine oocytes maturation in vitro. The outcome of this study serve as a reference point for the improvement of oocyte utilization, which in turn will benefit the related biotechnological applications.

## MATERIALS & METHODS

Unless otherwise stated, all reagents used in the present study were purchased from Sigma Chemicals (St. Louis, MO, USA).

### Oocyte collection and in vitro maturation

Porcine ovaries were procured from a local abattoir. The oocytes were collected from 3–6 mm diameter ovarian follicles by aspirating them with the 18-gauge needle attached to a disposable 20 mL syringe. We washed the oocytes four times with the Tyrode's lactate (TL)–Hepes–PVA (0.1%) and the compact Cumulus and Oocyte Complexes (COCs) were cultured in microdrops of maturation medium supplemented with 0, $10^{-6}$, $10^{-9}$, and $10^{-12}$ mol/L MT for in vitro maturation (IVM); 100 $\mu$L microdrop contained 30 COCs. All the groups were incubated at 39 ° C with 5% $CO_2$ in 95% humidified air for 42 h (*Huang et al., 2016*). The maturation medium, TCM199 (with Earle's Salts; Gibco), contained cysteine (0.1 mg/mL), penicillin (0.065 mg/mL), porcine follicular fluid (PFF)(10%), epidermal growth factor (EGF) (10 ng/mL), equine chorionic gonadotropin (eCG; Intervet Pty. Ltd, Australia) (10 IU/mL), and human chorionic gonadotrophin (hCG; Intervet Pty. Ltd.) (10 IU/mL).

### The assessment of oocyte maturation quality

After 42 h of maturation culture, COCs from each group were denuded by gentle pipetting in phosphate-buffered saline (PBS) supplemented with 0.1% hyaluronidase. The denuded oocytes from different groups were transferred to the TCM199 medium supplemented with 0.5% fetal bovine serum (FBS) and 25 mM Hepes (for washing). We marked the oocytes as survival oocytes after observing them for homogeneous cytoplasm and intact, bright membrane under a stereomicroscope (Fig. 1A); besides, the matured oocytes were the survival oocytes with the expulsed polar body into the perivitelline space (Fig. 1D). Survival and maturation of the oocytes were validated by the FDA and Hoechst33342 staining, respectively. The survived oocytes were characterized by the presence of bright fluorescence in the ooplasmic membrane (Fig. 1B); however, the absence of fluorescence indicated dead oocytes (Fig. 1C). The oocytes with the expulsed polar body exhibited nucleus and polar body fluorescence (Fig. 1E); however, the oocytes with no expulsed polar body showed fluorescence only in the nucleus (Fig. 1F). The matured oocytes were used for subsequent experimentations.

### Quantification of intracellular glutathione

As per the method reported by Huang et al., we performed a 5,5′-Dithiobis (2-nitrobenzoic acid) (DTNB)-GSH reductase recycling assay to determine the total intracellular concentration of GSH in a single oocyte of different groups (*Zhao et al., 2017*). A total of 20–30 oocytes from each group were frozen into a 1.5 mL centrifuge at $-80\,°C$ till we assayed the GSH content. The frozen oocytes were thawed and homogenized by repeated pipetting during the GSH content detection. Later, we transferred this homogenized solution into a 96-well plate and added a 150 mL assaying solution to each well. After the solution equilibration at 25 °C for 5 min, we added 50 mL of 0.16 mg/mL nicotinamide adenine

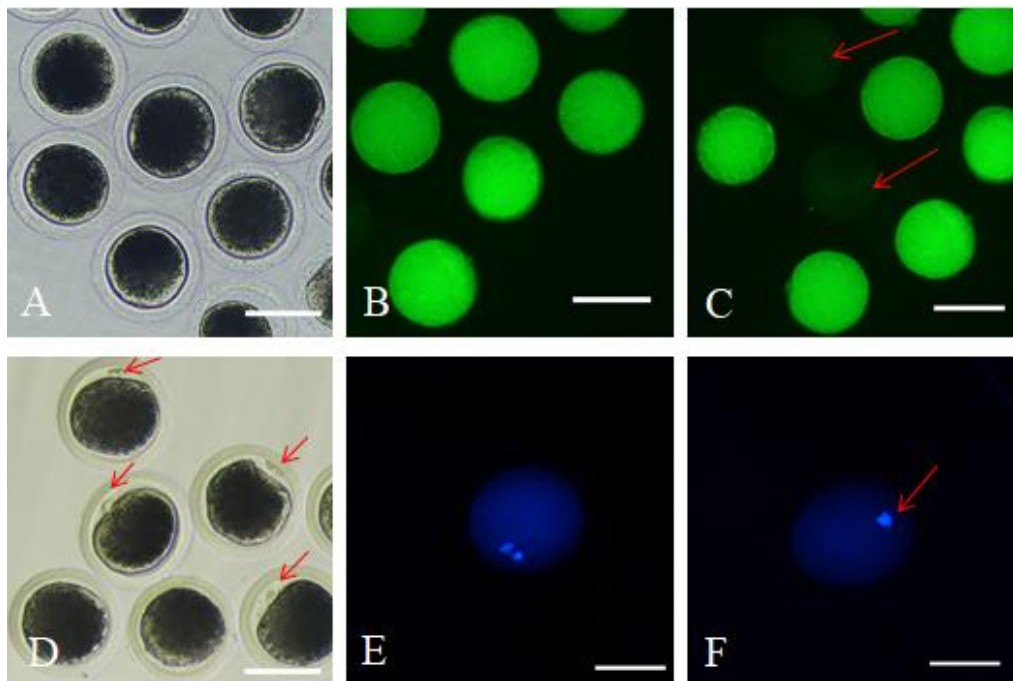

**Figure 1** **Survival and matured oocytes observed under stereomicroscopy and confirmed by fluorescence staining.** (A) Stereomicroscopic examination of oocytes. (B) Survival oocyte FDA staining validated. (C) Dead oocytes showed no or very low fluorescence as red arrow pointed. (D) The stereoscopic examination of the matured oocytes with the first polar body expulsed. (E) The matured oocytes stained with fluorescent dye Hoechst 33342 showed two fluorescent spots. (F) Oocytes with no polar body expulsed confirmed by fluorescent dye Hoechst 33342 only showed one fluorescent spot.

dinucleotide phosphate (NADPH) to each well, which led to the formation of 5-thio-2-nitro-benzoic acid. We measured the absorbance at 412 nm using a spectrophotometer (Beckman DU-40, USA) for five times at 30 s interval. The standard curve was plotted for the calculation of GSH value. We divided the value by oocytes number in each sample and detected a blank sample, i.e., without GSH, in our study.

## Mitochondrial distribution analysis

We washed one hundred oocytes from each group for four times in PBS containing 0.2 M sodium phosphate buffer. Oocytes were incubated in TCM-199 medium with 12.5 mmol/L Mito Tracker Red (Invitrogen, USA) at 37 °C under 5% $CO_2$ for 30 min and washed four times in PBS. Later, they were mounted on a slide and placed under a coverslip. The oocytes were observed under the fluorescence microscope (TE2000-s, Nikon, Japan). Two main distribution features characterized the porcine oocytes mitochondrial distribution pattern: homogeneous or even distribution (Fig. 2A) and heterogeneous or uneven distribution throughout the ooplasm (Figs. 2B–2E).

## Parthenote production and culture

A total of 80 oocytes from each group were transferred to the activation medium containing 1.0 mM $CaCl_2$, 0.1 mM $MgCl_2$, 0.3 Mm mannitol, and 0.5 mM HEPES

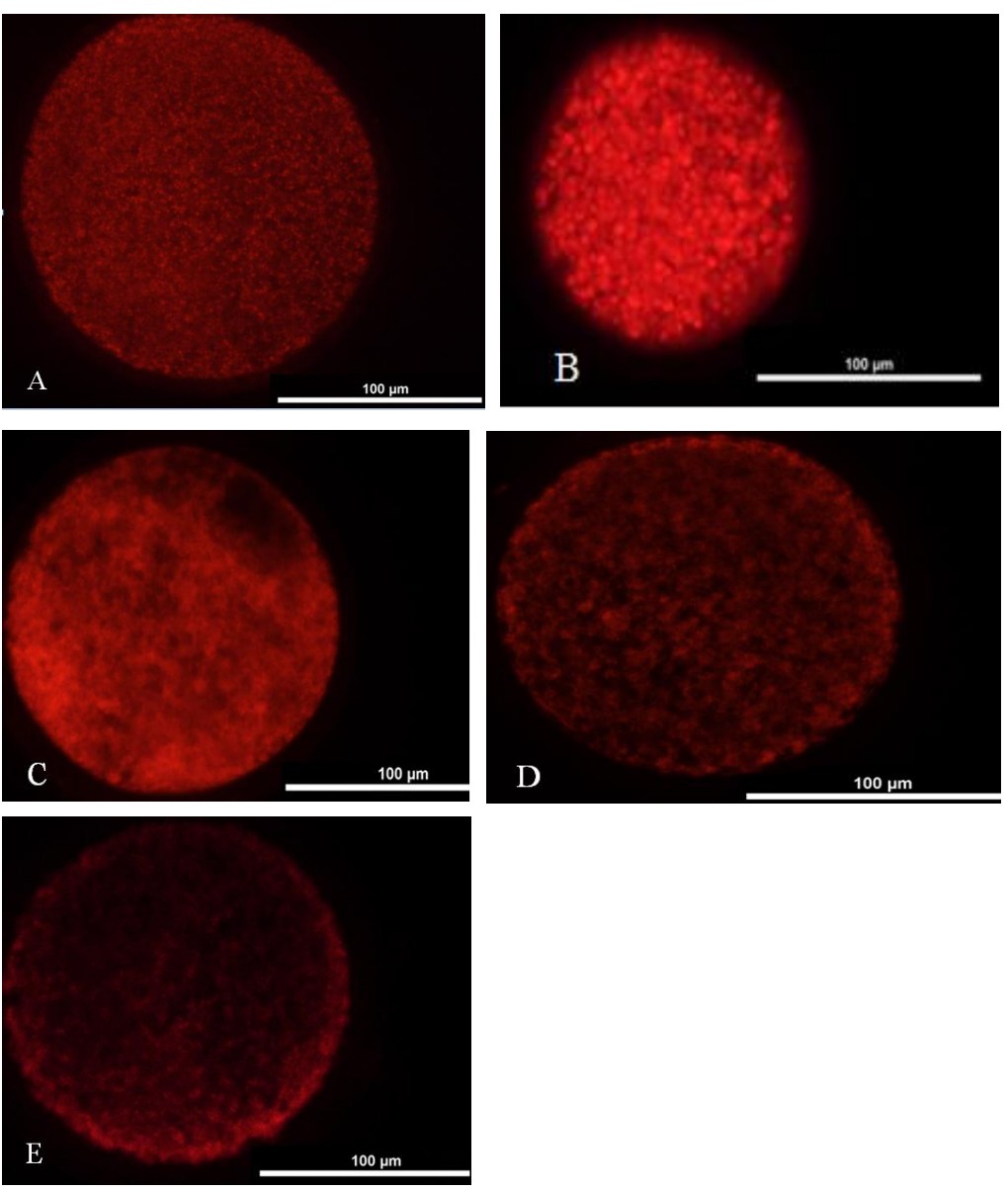

**Figure 2  Mitochondrial distribution status in matured oocyte.** (A) Oocyte with better maturation quality and even distribution of mitochondria in the ooplasm. (B–E) Oocytes with heterogeneous and uneven distribution of mitochondria in the ooplasm.

for the parthenogenetic activation (PA). Matured oocytes were activated with two pulses of 120 V/mm DC for 60 ms with the Electro-Cell Manipulator BTX 2001 (BTX Inc, USA). Subsequently, the parthenotes were cultured in 2 mM 6-dymethylaminopurine (6-DAMP) for 6 h, and later transferred into PZM-3 medium and incubated at 39 °C for 7 h at 5% $CO_2$ (*Huang et al., 2016*). We observed cleavage and blastocyst formation on days 2 and 7, respectively, after oocytes activation.
**Table 1 The primers and Genebank source accessions for each gene used in the study.** RT-PCR primers and Genebank source.

| Gene | Primer sequence (5′–3′) | Accession number |
|------|--------------------------|------------------|
| GAPDH | F:TCAAATGGGGTGATGCTGGT<br>R:GCAGAAGGGGCAGAGATGAT | XM_021091114 |
| MAPK | F:AGTGCCTACCATGCTTCTCG<br>R:TTGTGGTTGTCCTCAACGGG | XM_021071922 |
| MPF | F:ACTGGCTAGTGCAGGTTCAG<br>R:TTGGAGCATCTTCTTGGGCA | XM_003124695 |

## Detection of gene expression with real-time polymerase chain reaction (RT-PCR)

RT-PCR was used to detect the expression activity of MAPK and MPF genes. RNA was extracted from a total of 100 oocytes from each group by using TRIzol reagent (Invitrogen) as per the manufacturer's instructions. cDNA synthesis was performed for 30 min at 55 °C using Omniscript reverse transcription Kit (Invitrogen) with oligo-dT primer, and PCR was performed by using the Maxime PCR Premix with SYBR Green (TaKaRa Bio Inc., Otsu, Japan). The PCR reaction mixture contained specific primers for cDNA samples. The cDNA was amplified under the following conditions: predenaturation at 95 °C for 3 min, denaturation at 95 °C for 15 s, annealing at 56 °C for 30 s, elongation at 72 °C for 30 s, and final extension at 72 °C for 5 min for 40 cycles using Eppendorf Mastercycler (Eppendorf, Hamburg, Germany). We designed primers with Primer 5.0 software on the basis of the mRNA sequences of Sus scrofa genes published in GenBank, which were synthesized by Shanghai bioengineering co., LTD (Shanghai, China). The primers used in the present study had been verified to be available by RT-PCR. Real-time quantitative PCR data were analyzed by employing the comparative Ct ($2^{-\triangle\triangle Ct}$) method, and the relative expression level of each gene from each cDNA pool was normalized against the reference gene GAPDH. PCR amplification efficiency of each pair of primers was assessed before quantification, and was found to be in an acceptable range (between 0.9 and 1.1).

The primers and Genebank source accessions for each gene are reported in Table 1.

## Statistics

We performed the log transformation of the percentage values before the analyses. The quantitative data were analyzed by least-squares ANOVA using the General Linear Models (GLM) procedures of the Statistical Analysis System (SAS, version 9.4) (Institute, Cary, NC, USA). We corrected Real-time PCR data by using the GAPDH data as a covariate for different analyses. All data were expressed as mean ±SEM, and different letters, such as a, b,or c over a bar or a column was considered as statistically significant ($p < 0.05$). All experiments were repeated thrice.

**Table 2** The effect of melatonin on the qualities of porcine oocytes maturation in vitro.

| MT concentration (mol/L) (mol/L) | Total oocytes | Rate of survival oocytes (%) | Rate of matured oocytes (%) |
|---|---|---|---|
| 0 | 200 | $89.67 \pm 1.86^a$ | $78.00 \pm 0.58^a$ |
| $10^{-12}$ | 200 | $89.33 \pm 1.48^a$ | $80.50 \pm 0.76^b$ |
| $10^{-9}$ | 200 | $91.33 \pm 1.97^a$ | $85.50 \pm 1.26^c$ |
| $10^{-6}$ | 200 | $91.67 \pm 1.17^a$ | $81.33 \pm 1.17^b$ |

**Notes.**
Within a column, percentages with a common superscript mean no significant difference ($p > 0.05$), and with different superscript (a–c) means a significant difference ($p < 0.05$). Each experiment was repeated thrice.

**Table 3** The GSH content in a single oocyte of different groups.

| MT concentration (mol/L) | Oocyte numbers | GSH content (pmol/oocyte) |
|---|---|---|
| 0 | 100 | $8.25 \pm 0.13^a$ |
| $10^{-12}$ | 100 | $9.25 \pm 0.08^b$ |
| $10^{-9}$ | 100 | $9.34 \pm 0.05^b$ |
| $10^{-6}$ | 100 | $8.99 \pm 0.07^b$ |

**Notes.**
Within a column, the percentage with different superscript (a–b) means a significant difference ($p < 0.05$). Each experiment was repeated thrice.

# RESULTS

## Effects of melatonin on survival and maturation rate after porcine oocytes maturation culture in vitro

As shown in Table 2, no significant difference in survival rate was observed among the four groups ($p > 0.05$), however, the maturation rates in the three experiment groups were all significant higher than the Control, and the maturation rate in $10^{-9}$ mol/L group was significant higher than the $10^{-6}$ and $10^{-12}$ group ($p < 0.05$). There was no significant difference existed between $10^{-6}$ and $10^{-12}$ group ($p > 0.05$). Figures 1A, and 1D represents the survival and matured oocytes, respectively. Figures 1B, 1C, 1E and 1F show that the survival and mature oocytes were confirmed by fluoresent staining.

## The effect of melatonin on glutathione content of single oocyte in different groups

Table 3 shows that the GSH content of a single oocyte in three experimental groups was significantly higher than the control group ($p < 0.05$); Although the GSH content in $10^{-9}$ mol/L group was the highest, there was no significant difference among the three experimental groups ($p > 0.05$).

## Mitochondrial maturation distribution of porcine oocytes in different groups

Figure 3 shows that the proportions of oocytes with better mitochondrial distribution in the $10^{-12}$, $10^{-9}$ and $10^{-6}$ mol/L group were all significantly higher than that in the 0 mol/L group, and the proportion in $10^{-9}$ mol/L group was significantly higher than that in the $10^{-12}$ and $10^{-6}$ mol/L group ($p < 0.05$), whereas $10^{-12}$ mol/L group had a

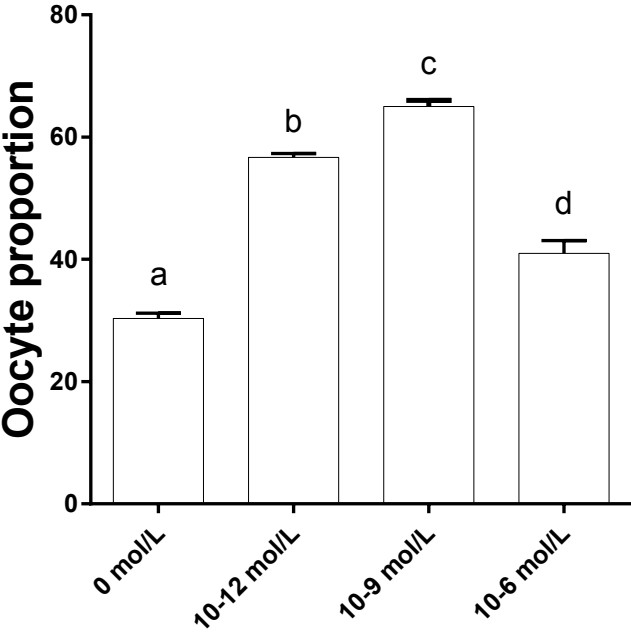

**Figure 3  The proportion of oocyte with better mitochondrial distribution in different groups.** The proportion of oocyte with better mitochondria distribution. Different lowercase letters (a–d) over a bar mean significant difference ($p < 0.05$). Each experiment was repeated three times.

significantly higher proportion of oocytes with better mitochondrial distribution than that in the $10^{-6}$mol/L group ($p < 0.05$). Figure 2 represents the status of mitochondrial maturation distribution.

## To study the developmental potential of the parthenogenetic embryo in different groups

As illustrated in Fig. 4 the cleavage rate in the $10^{-6}$ mol/L and $10^{-9}$ mol/L groups were significantly higher than those in the 0 and $10^{-12}$ mol/L group ($p < 0.05$). Similarly, the blastocyst rate in the $10^{-6}$ mol/L and $10^{-9}$ mol/L groups were also significantly higher than those in the 0 and $10^{-12}$ mol/L group ($p < 0.05$). However, we did not observe a significant difference in the cleavage and blastocyst rate between the $10^{-6}$ mol/L and $10^{-9}$ mol/L groups and the 0 and $10^{-12}$ mol/L group ($p > 0.05$).

## The effects of melatonin on the mRNA expression of MPF and MAPK genes in oocytes

Figure 5 demonstrates that the mRNA expression levels of MPF and MAPK genes in the $10^{-9}$ mol/L group were significantly higher than the other three groups ($p < 0.05$). The MPF gene expression level of $10^{-6}$ mol/L group was higher than those in the 0 and $10^{-12}$ mol/L group ($p < 0.05$), however, no significant difference existed between the 0 and $10^{-12}$ mol/L group. There was no significant difference in MAPK gene expression between the

Peer

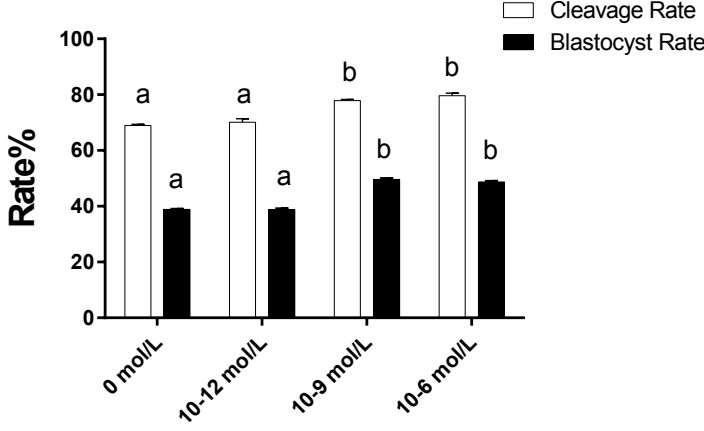

**Figure 4  Cleavaged embryos and blastocyst rate in different groups.** Developmental potential of the embryos in different groups. Different lowercase letters (a–b) over a bar of cleavage rate or over a bar of blastocyst rate represent a significant difference ($p < 0.05$). Each experiment was repeated thrice.

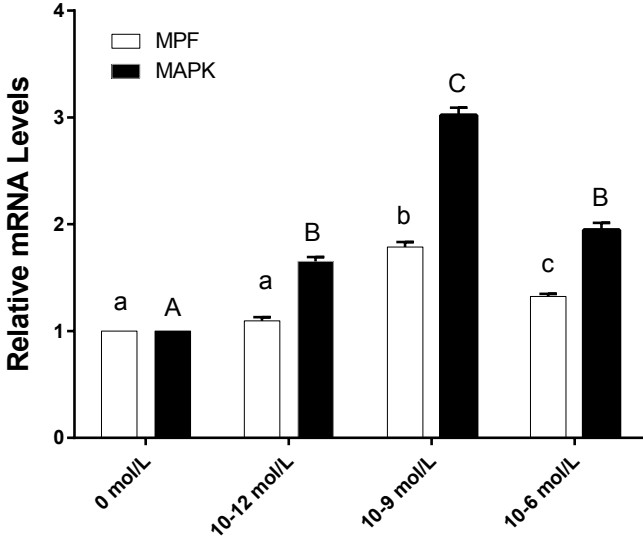

**Figure 5  The mRNA expression of MPF and MAPK genes in ooyctes of differnt groups.** MPF and MAPK mRNA expression in different groups. Different lowercase letters (a–c) over a bar of MPF expression or different uppercase letters (A–C) over a bar of MAPK expression represent a significant difference ($p < 0.05$). Each experiment was repeated thrice.

$10^{-6}$ and $10^{-12}$ group ($p > 0.05$), whereas, they were both significantly higher than the 0 mol/L group ($p < 0.05$).

## DISCUSSION

Previous studies have shown the beneficial effect of melatonin on the in vitro maturation (IVM) of porcine oocytes (*Li et al., 2015*), embryo development (*Choi et al., 2008*), and the in vitro protection of fertilized embryos (*Rodriguez-Osorio et al., 2007*). We did not find a significant influence on porcine oocyte survival after oocyte had been maturation cultured with different concentration of melatonin for 42 h in vitro. We attribute this results to the ideal maturation conditions established in our lab, where the culture environment was suitable to meet the requirements of oocyte's energy, nutrients, and other maturation demands (*Huang et al., 2016*). As shown in Table 2, the lowest survival rates of the four groups was 88.94%. However, the maturation rate was significantly increased in the $10^{-6}$, $10^{-9}$ or $10^{-12}$ mol/L group comparing to the 0 mol/L group. Because the process of oocyte maturation involves many molecular events and biochemical reactions, which coordinating nuclear and cytoplasmic maturation (*Arias-lvarez et al., 2017*), so our study demonstrated that $10^{-12}$–$10^{-6}$ mol/L melatonin could further promote oocyte maturation in vitro. Besides, we found that different melatonin concentrations can elevate the porcine oocyte maturation quality, as indicated by the single oocyte's GSH content and the higher ratio of the oocyte with better mitochondrial distribution, especially, $10^{-9}$ mol/L melatonin was found to be the most effective concentration in the present experiments. Previous studies suggest that melatonin acts as an antioxidant and protects the oocytes from the adverse effect of ROS during in vitro maturation. Thus, it promotes oocyte's maturation and developmental ability (*Shi et al., 2009*). Melatonin significantly improves the oocytes cytoplasmic maturation by improving the ratio of oocytes with the normal distribution of organelles and by increasing the intracellular GSH and ATP levels (*Zhao et al., 2017*), which were consistent with our study that melatonin could significantly improve the mitochondrial maturation distribution in cytoplasm. In the recent study by our group, we found that melatonin concentration of $10^{-5}$ M significantly improved the quality of mitochondria maturation in porcine oocyte as compared to the control group. However, these beneficial effects of melatonin could be blocked by $10^{-5}$ M luzindole, which is a melatonin receptor antagonist (*Yang et al., 2020*). As per the previous report, the melatonin concentrations of $10^{-3}$–$10^{-11}$ M positively affects the porcine oocyte maturation, and $10^{-9}$ M melatonin is the optimum concentration for porcine oocyte maturation in vitro (*Shi et al., 2009*). The outcomes of our analysis were in line with this study.

The physiological dose of ROS plays an important role in cell growth and metabolism (*Rajani et al., 2012*) and the enhanced of ROS content in cells can induce DNA damage and lipid peroxidation, disrupt mitochondrial function (*Loven, 1988*; *Lord-Fontaine & Averill-Bates, 2002*). We speculate that $10^{-9}$ mol/L melatonin keeps the ROS content at an appropriate level and protect the oocytes from oxidative stress damage, thereby maintaining the physiological dose of ROS to support the ooplasmic maturation of the porcine oocytes in vitro. From the perspective of the embryo cleavage and blastocyst rate, they were both significantly higher in $10^{-6}$ mol/L and $10^{-9}$ mol/L groups against the 0 and $10^{-12}$ mol/L group. These results might be correlated with the better maturation quality of porcine oocytes. We had already shown that $10^{-6}$ mol/L and $10^{-9}$ mol/L groups exhibit a higher

GSH content in single oocyte and a higher ratio of the oocyte with better cytoplasmic mitochondrial distribution, which were essential for embryo development (*Huang et al., 2016*). Numerous studies have reported that melatonin, an antioxidant, scavenge ROS (*Tamura et al., 2012*), and promotes porcine oocyte maturation as well as embryonic development (*Choi et al., 2008*; *El-Raey et al., 2011*; *Lord et al., 2013*; *Tamura et al., 2009*; *Tian et al., 2014*). *Li et al. (2015)* also tested the concentration of $10^{-6}$ mol/L-$10^{-9}$ mol/L melatonin to the influence of oocyte maturation and development when oocytes were heat stressed, in which $10^{-9}$ mol/L melatonin exhibited the best protective effects from heat stress and boosted porcine oocytes maturation and development. These studies are in line with our experimental outcomes.

The MPF and MAPK genes play an essential role in the maturation and development of oocytes (*López-Cardona et al., 2017*). Other studies also illuminated the fact that the MPF gene plays a central role in oocyte maturation and embryonic development by regulating oocytes meiosis and cell cycle; also, it promotes nuclear maturation of porcine oocytes (*Dadashpour Davachi et al., 2017*; *Liu et al., 2018*; *Oqani et al., 2017*). In our study, the expression pattern of the MPF gene varied with the changes of oocyte maturation quality and developmental potential, and the supplement of melatonin with $10^{-9}$ mol/L is most beneficial to porcine oocytes maturation and development during maturation in vitro, while the higher concentration with $10^{-6}$ mol/L or the lower concentration with $10^{-12}$ mol/L melatonin supplement both decreased the expression of the MPF gene. Nevertheless, another report showed that the most suitable concentration of melatonin for porcine oocyte maturation and development was $10^{-7}$ mol/L (*Li et al., 2015*). Li et al. studied the porcine oocytes under heat stress conditions (42 °C for 20–24 h during IVM). The findings of this study suggested that $10^{-7}$ mol/L melatonin protected and promoted the oocyte maturation and development under heat stress condition; however, in the normal IVM condition, $10^{-9}$ mol/L melatonin promoted the maturation and development of the oocytes. Previous studies suggested that a high MAPK gene expression activity is an essential marker of oocyte maturation and a necessary criterion for the oocyte's maturation quality (*Li et al., 2017*; *Sun et al., 2016*). Melatonin can activate the MAPK protein and regulate the MPF protein by interacting with intracellular transcription factors or cell inhibitory factor (CIF), which control the meiosis of oocyte and promote oocyte maturation and embryo development (*Mayo et al., 2016*; *Tiwari et al., 2017*). However, in our present study, the MAPK gene expression level was showed significantly higher in $10^{-9}$ mol/L goup than the other groups ($p < 0.05$), which was also consistent with the results of maturation indexes and MPF gene expression. The outcome of the current study shows that the group with $10^{-9}$ mol/L group melatonin supplement exhibits the highest maturation quality and developmental potential consistent with the heightened MPF and MAPK mRNA expression.

## CONCLUSION

Based on our previous findings, in the current study, we reported the optimum melatonin-supplement concentration ($10^{-9}$ mol/L) for porcine oocyte maturation culture at 39 °C, 5% $CO_2$, 95% humidity for 42 h in the vitro condition. We found increased single oocyte

GSH content, better mitochondrial maturation distribution, maturation rate, MPK, and MAPK gene expression in the $10^{-9}$ mol/L group, which increased the developmental potential of oocytes. The outcome of the current study extended our understanding of melatonin-induced porcine oocyte maturation and embryo development, which has provided a reference point for the biotechnological applications of oocyte maturation and development in vitro.

### Funding

This work was supported by the National Key Research and Development Program of China (2016YFD0500502), The Special Fund for Pig Industry Technology Innovation Team of Modern Agricultural Industry Technology System in Tianjin City (ITTPRS2017001) and The Tianjin "131" Innovative Talents Team (20180338). The funders had no role in study design, data collection and analysis, decision to publish, or preparation of the manuscript.

### Grant Disclosures

The following grant information was disclosed by the authors:
National Key Research and Development Program of China: 2016YFD0500502.
Special Fund for Pig Industry Technology Innovation Team of Modern Agricultural Industry Technology System in Tianjin City: ITTPRS2017001.
The Tianjin "131" Innovative Talents Team: 20180338.

### Competing Interests

The authors declare there are no competing interests.

### Author Contributions

- Zimo Zhao performed the experiments, analyzed the data, prepared figures and/or tables, and approved the final draft.
- Ling Yang and Qianjun Li conceived and designed the experiments, authored or reviewed drafts of the paper, and approved the final draft.
- Dan Zhang analyzed the data, prepared figures and/or tables, and approved the final draft.
- Zi Zheng and Ning Li performed the experiments, prepared figures and/or tables, and approved the final draft.
- Maosheng Cui conceived and designed the experiments, analyzed the data, prepared figures and/or tables, authored or reviewed drafts of the paper, and approved the final draft.

### Data Availability

The raw data of statistical results, the raw pictures of the survival oocytes, the raw pictures of the oocytes with or without the expulsed first polar body, the raw pictures of the mitochondrial distribution status in ooplasm, and the raw pictures of the embryo cleavage and blastocyst are available as Supplemental Files.

## Supplemental Information

Supplemental information for this article can be found online at http://dx.doi.org/10.7717/peerj.9913#supplemental-information.

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
