# Peer review of "Elevation of MPF and MAPK gene expression, GSH content and mitochondrial distribution quality induced by melatonin promotes porcine oocyte maturation and development in vitro"

_PeerJ, doi:10.7717/peerj.9913_

## Round 0.1 · original submission · Major Revisions

Dear Dr. Cui,

As stated by both reviewers, the manuscript cannot be accepted for publication in its current form. An extensive number of experiments must be provided, particularly to support the title and the claims about the melatonin effects into MPF and MAPK gene expression, GSH content and mitochondrial distribution quality. Extensive rewriting is required. It should be clarified the relationship between the current data and previous manuscripts from the group, which should be properly cited. Please provide new experiments of qRT-PCR as suggested by Reviewer #2. In addition, changes in mitochondrial distribution quality cannot be observed with the current panels present in the manuscript. Please also follow all the suggestions of Reviewer #1 particularly regarding graphical statistical representation and to provide in the discussion a possible explanation of why a higher concentration of Melatonin is not necessarily the best condition supporting oocyte health and maturation.

Best regards,
Rodrigo

Reviewer 1 ·

Basic reporting

The current study by Zhao et al. demonstrates the beneficial effect of melatonin supplementation on porcine oocyte maturation in vitro. They report biochemical changes due to melatonin treatment including changes in GSH levels and mitochondrial distribution which are known to be beneficial for oocyte quality and fertilizability and verify the beneficial effect on oocyte developmental potential. Finally, they show that melatonin (particularly at a concentration of 10-9 mol/L) treatment correlates with significant elevation in the expression of MPF and MAPK, both of which are critical for oocyte meiotic progression and maturation. The important conclusion from the study is that 10-9 mol/L Melatonin supplementation is an ideal condition for in vitro maturation of porcine oocytes, an insight that can be of relevance to the animal biotechnology industry.

The study is appropriately designed and amply supported by references to existing literature in the field. One potential area for improvement is the quality of the writing. It requires significant editing to get rid of errors and to make the article more comprehensible to the reader. Examples: the use of words such as ‘furtherly’, ‘ovary maturation’ instead of oocyte maturation in first sentence of introduction, very long single sentences (e.g. line 26-32), and multiple typos and errors all through the text (e.g. line 194: the word ‘difference’ is missing). Also, the abstract itself is very long and repetitive – it would definitely help to have a simpler and more compact abstract.

Experimental design

The research question has been suitably defined and it draws a specific conclusion about the specific concentration of Melatonin which is most beneficial for maturation of porcine oocytes.

Validity of the findings

The beneficial effects of Melatonin described are correlational and the authors rightfully state this in the text. The findings are within the scope of the study, appropriately stated and supported by the experimental results and statistical tests.

Additional comments

Major comments:
Even though the authors have extensively mentioned existing literature and context for their work, it’s important to note that they have not cited Yang et al., 2020 (Effect of Melatonin on the In Vitro Maturation of Porcine Oocytes, Development of Parthenogenetically Activated Embryos, and Expression of Genes Related to the Oocyte Developmental Capability). This recent paper appears to be from the same (or related) research group with three authors common to both studies. This is important because, the previous study and results overlaps to some degree (beneficial effect of melatonin on GSH levels, mitochondrial distribution and parthenogenetic activation) with the current manuscript. Keeping that in mind, the two main findings exclusive to the new report are the effects on MAPK and MPF expression and the recommendation of the most beneficial Melatonin concentration of 10-9 mol/L.

Minor comments:
1. The current representation of statistical significance in the histograms and figures is very unconventional, using dissimilar/similar letters to denote significance/the lack of it. While this is not incorrect and serves the purpose, it can be a bit confusing to understand which conditions are being compared. A brief key is provided in the figure legends, but it will be better if the authors can include a couple of sentences explaining this in the ‘statistics’ section of Materials and Methods.

2. It would be nice if the authors would include in the discussion their view about why a higher concentration of Melatonin (10-6 mol/L) is not necessarily the best condition supporting oocyte health and maturation. One would expect a dose-response where the beneficial effects correlate directly with the dose of Melatonin. It’s possible that excessive Melatonin could activate feedback mechanisms that interact with other pathways thereby reducing effectivity.

Reviewer 2 ·

Basic reporting

MPF and MAPK pathway have many genes. The author only analyzed the expression of one gene in this pathway, so the results have·low credibility. Just verifying the related single gene by PCR does not show that melatonin could affect oocytes through MPF and MAPK pathways

Experimental design

The overall experimental design of the paper is very simple and lack of logic. The quality of the figures is poor, and the figures cannot support the results of the paper.

Validity of the findings

The results of the paper lack innovative. There are many papers published on related research results (PMID: 32012669, 31773776, 31672358). This paper just repeated the relevant experimental results, but they are relatively low. Only the cleavage rate and blastocyst rate are not enough to explain the effect of melatonin on oocyte development

Additional comments

In general, the lack of innovation is serious, and the results 1234 are just repeating the results of previous studies. At the same time, there is a lack of sufficient data to support the problem that the experiment wants to explain. In addition, the coherence and logic of the paper are poor, and no related channel issues involved in the title are raised. This paper cannot meet peerJ's publishing requirements.

---

## Round 0.2 · Major Revisions

Dear Dr. Cui,
Please follow all the required changes suggested by single reviewer #1. Particularly important is the requirement of better quality and additional images for Figure 1 and 2. Without these experiments the paper cannot be accepted since the statements in the text must be supported by the provided data.

Reviewer 1 ·

Basic reporting

The revised version of Zhao et al has improved from the original submission in several aspects. First, it has now included a highly relevant citation from recent literature and discussed their latest findings in context. The readability of the paper is also now significantly improved from the original version, thanks to language editing.

Experimental design

N/A as no new experiments were performed for this revision.

Validity of the findings

The authors have provided the raw data based on which they performed statistical tests and prepared their figures. They have also included a sentence about the style in which significance is assessed in their assays per the previous request.

As mentioned before, the beneficial effects of melatonin described are correlational as the authors only test the expression changes of two genes in a complex cascade. This is not necessarily problematic as the data is reported and presented as correlative, focusing instead on specific phenotypic effects of melatonin stimulation in combination with the gene expression analysis.

Additional comments

Potential area of improvement: The resolution of the images is really poor and it's possible that it is a consequence of the pdf generation process during or after submission. I would certainly recommend having images of much better resolution for the publication (Figure 1 and 2).

For example, from Fig 2, it is hard to see any detail in the mitochondrial network due to excessive signal intensity and poor resolution. Also, rather than just one example each of 'good' and 'bad', I strongly recommend:
a) Showing images from multiple oocytes for each melatonin concentration
b) Showing the range of phenotypes, as can be better appreciated from the original figures shared with us but not incorporated into manuscript.

Please make sure to include scale bars in all images for proper comparison. It looks like they may not have been acquired at the same magnification. Also, please make sure that the signal intensity scaling is done identically across all conditions, while preparing the figures.

---

## Round 0.3 · accepted · Accept

Dear Dr. Cui,

I would like to congratulate you for improving the images. Reviewer 1 was satisfied with the improvement, thus after two rounds of revisions, the manuscript is accepted.

Reviewer 1 ·

Basic reporting

N/A

Experimental design

N/A

Validity of the findings

N/A

Additional comments

Thank you for improving the quality of the figures and adding in details for clarity.